# Biomimetic Hydrofoil Propulsion: Harnessing the Propulsive Capabilities of Sea Turtles and Penguins for Robotics

**DOI:** 10.3390/biomimetics10050272

**Published:** 2025-04-28

**Authors:** Yayi Shen, Zheming Ding, Xin Wang, Zebing Mao, Zhong Huang, Bai Chen

**Affiliations:** 1College of Mechanical and Electrical Engineering, Nanjing University of Aeronautics and Astronautics, Nanjing 210016, China; yayi.shen@gmail.com (Y.S.); zheming.ding.102@gmail.com (Z.D.); xwang381@163.com (X.W.); 2Yangtze River Delta Intelligent Manufacturing Innovation Center, Nanjing 210004, China; 3Department of Engineering Science and Mechanics, Shibaura Institute of Technology, 3-7-5 Toyosu, Koto-ku, Tokyo 135-8548, Japan; mao.zebing.v.5@sic.shibaura-it.ac.jp; 4School of Information and Communication Engineering, Hainan University, Haikou 570228, China; zhonghuang@hainanu.edu.cn

**Keywords:** flapping hydrofoil, biomimetics, underwater robot, sea turtle forelimb, penguin wing

## Abstract

This review synthesizes current research on hydrofoil-propelled robots inspired by the swimming mechanisms of sea turtles and penguins. It begins by summarizing the swimming kinematics of these organisms, highlighting their superior aquatic performance as the primary motivation for biomimetic design. Next, established analytical methods for characterizing hydrofoil locomotion patterns are presented, along with a clear delineation of the decoupled motion components exhibited by sea turtle flippers and penguin wings. Such decoupling provides a systematic framework for guiding the design of driving mechanisms. Building on this biomechanical foundation, the review critically examines recent advances in biomimetic flexible hydrofoils that enhance propulsion efficiency through three synergistic mechanisms to enhance thrust generation, while identifying key challenges in material durability and non-linear fluid–structure interactions. The review then surveys existing hydrofoil actuation systems, which commonly reproduce coupled motions with multiple degrees of freedom (DOFs). Finally, representative biomimetic robots are examined: sea turtle-inspired forelimbs typically incorporate three DOFs, whereas penguin-inspired wings usually offer two DOFs. By aligning robotic designs with the decoupled motion patterns of the source organisms, this review offers critical insights to advance the development of hydrofoil propulsion systems for enhanced aquatic performance.

## 1. Introduction

As human civilization advances, the strategic value of ocean resources is becoming increasingly prominent. Oceans cover 70% of the Earth’s surface and are rich in natural resources. To fully exploit the vast potential of the ocean, robots designed for underwater operations have become essential. Underwater robots are highly promising intelligent devices with broad application prospects in areas such as environmental monitoring, resource exploration, search and rescue, and military applications. Over the past several decades, various types of underwater robots have been developed, playing key roles in different application scenarios [1,2,3,4]. Biomimetic underwater robots are a class of robots that generate thrust by mimicking the propulsion methods of underwater animals or amphibians in nature. Compared to traditional underwater robots that use axial propellers to generate thrust, biomimetic underwater robots offer advantages in terms of energy efficiency, maneuverability, safety, and noiseless motion [5].

The swimming methods of biological organisms determine the propulsion mechanisms of the corresponding biomimetic robots. Swimming methods in the animal kingdom can be primarily classified into four types: swaying, rowing, hydrofoil, and jetting [6]. One swimming method that has been extensively studied is the swaying swimming method, which is used by fish in nature [7,8]. In Lighthill’s elongated-body theory, the traveling-wave equation is used to describe the midline curvature of fish bodies [9]; recently, Yong et al. proposed a general kinematic model, composed of a non-linear oscillator and a traveling-wave equation, which can demonstrate all the aforementioned swimming motions of the robotic fish [10]. Robots utilizing body and/or caudal fin (BCF) propulsion modes [11,12,13,14,15], as well as median and/or paired fin (MPF) propulsion modes [16,17,18,19,20], along with robots that combine both propulsion methods [21,22,23], have been widely researched with significant progress. Among these, BCF-mode robots, which mimic tuna or dolphins and generate thrust through body or tail undulation, generally exhibit superior propulsion performance, with higher speeds and excellent acceleration. However, these robots are more challenging to control and tend to have poorer stability during movement. MPF mode robots, which mimic stingrays or eels and generate thrust through the undulation of fins along the body or at the sides, trade off lower speeds for more precise attitude control, exhibiting greater stability in movement compared to BCF-mode robots [24].

The application potential of undulatory swimming in the field of biomimetic robots has been confirmed. Among the remaining three swimming methods, rowing swimming (as seen in amphibians such as frogs) is considered to have lower swimming efficiency. Rowing swimming is a drag-based propulsion method, characterized by a long recovery phase during each movement cycle. The presence of this recovery phase prevents continuous thrust generation during one movement cycle. This intermittent propulsion reduces overall propulsion efficiency. Biomimetic robots based on rowing propulsion are commonly seen in surface vehicles, where the low density of air helps mitigate the hydrodynamic drag caused by the recovery phase [25]. Jet propulsion swimming (as seen in soft-bodied marine creatures like squid) expels water through muscular contraction, using the reaction force of the water to propel the animal forward. This swimming method is energy-intensive and cannot be sustained for long periods. In the field of biomimetic robots, jet propulsion is often used in robots employing hybrid propulsion modes, where additional thrust is obtained for a short duration with high energy consumption [26].

The final swimming method is hydrofoil propulsion. Hydrofoils refer to the forelimbs of certain marine animals, such as the forelimbs of sea turtles and penguins [27]. The hydrofoil has a good streamline structure along the chord-wise section in the direction of the incoming flow, which can significantly reduce the resistance caused by the hydrofoil dynamic swirl [28]. The developmental histories of penguin robots and sea turtle robots are illustrated in Figure 1. Additionally, the swimming methods of sea lions and alcids also feature elements of hydrofoil propulsion [29]. The hydrofoils in these marine animals are similar to the wings of air birds, capable of performing multiple degrees of motion, such as flapping, feathering, and sweeping, simultaneously. Hydrofoil propulsion is a lift-based propulsion method, which means that the thrust during swimming is derived from the component of hydrodynamic lift in the forward direction, rather than the counterthrust generated by hydrodynamic drag [30]. The complete movement cycle of the hydrofoil can be roughly divided into an upstroke (from ventral to dorsal movement) and a downstroke (from dorsal to ventral movement), based on the direction of movement along the back and belly sides. The movement of the hydrofoil involves the coupling of large-amplitude flapping motion and smaller-amplitude movements (such as feathering and sweeping). Both the upstroke and downstroke can generate thrust [30,31], while the transient recovery phase between the upstroke and downstroke occupies only a small portion of the movement cycle. As a result, thrust can be considered continuously generated. This propulsion method thus boasts high propulsion efficiency. In addition to high propulsion efficiency, robots utilizing hydrofoil propulsion also have significant potential for achieving efficient motion control. The ideal biomimetic hydrofoil driving mechanism can generate large-amplitude, high-frequency, multi-degree-of-freedom coupled motions. Independent control of the hydrofoils on each side of the body provides the possibility for agile swimming of the robot. Moreover, due to the unique motion characteristics of the hydrofoil mechanism, hydrofoil-propelled robots exhibit a variety of locomotion patterns that distinguish them from robots using other propulsion methods. In contrast to conventional oscillatory or undulatory propulsion systems, which typically rely on frequency modulation to switch between high-thrust and low-power states, the hydrofoil mechanism offers a more efficient and adaptable transition between these modes by leveraging the interaction of lift and drag forces at varying angles of attack (AOAs) (the angle of attack(AOA) typically refers to the angle between the incoming flow and the chord line of the hydrofoil). This allows for a more smooth and continuous adjustment between high-thrust acceleration and low-power gliding, which can result in improved energy efficiency over a broader range of operational conditions. Finally, the diverse locomotion patterns enable hydrofoil-propelled robots to excel in the field of multi-medium robotics, extending beyond underwater operations. For instance, biomimetic robots based on turtles can transition between walking, crawling, and swimming states by switching the motion modes of their forelimbs, allowing movement across various media [32]. Similarly, alcids, capable of both swimming and flying, provide biomimetic inspiration for the design of multi-medium flapping-wing vehicles, based on their morphing wings [33].

Many recent studies have reviewed the progress of underwater biomimetic robots, with most of them focusing on robots mimicking fish. Yu et al. summarized the work on the motion control and motion coordination of robotic fish [34]. Wang et al. provided a comprehensive review of biomimetic underwater robots, focusing on BCF (i.e., body and/or caudal fin), MPF (i.e., median and/or paired fin), and hybrid locomotion modes, as well as motion control methods [35]. Li et al. summarized biomimetic robots inspired by fish [8]. Sun et al. explored methods for simulating fish characteristics in engineering applications [7]. Li et al. reviewed the swimming methods of fish and undulatory propulsion robots [36]. These works cover most mainstream biomimetic fish robots. However, underwater robots utilizing hydrofoil propulsion are rarely addressed. This paper attempts to provide a comprehensive review of hydrofoil propulsion based underwater robots, from the swimming methods of biological organisms to the implementation of robot driving mechanisms. The aim is to help researchers quickly familiarize themselves with and understand the principles of hydrofoil propulsion, draw inspiration from existing research, and use it to achieve breakthroughs.

The remainder of this paper is organized as follows: Section 2 provides a detailed introduction to the swimming methods of hydrofoil propulsion, Section 3 presents the implementation of driving mechanisms for hydrofoil propulsion, Section 4 summarizes and reviews existing hydrofoil propulsion robots, and Section 5 discusses the development trends of hydrofoil propulsion robots.

## 2. Hydrofoil-Based Swimming in Biology

The biological organisms in nature that use hydrofoil propulsion for swimming mainly include sea turtles and penguins. As typical representatives of hydrofoil propulsion, these two species share similarities in their swimming movements. Research into the swimming methods of sea turtles and penguins has made varying degrees of progress. Researchers have analyzed the swimming processes of these animals through observational experiments and recorded their kinematics. These kinematic studies provided important references for the development of hydrofoil propulsion robots, including the design of driving mechanisms, flexible hydrofoils, and streamlined shells. This section will introduce the research progress on the kinematics of sea turtles and penguins. The characteristics and advantages of the hydrofoil propulsion swimming method will also be summarized and analyzed.

### 2.1. Hydrodynamics of Sea Turtle Forelimbs

The body of a sea turtle is constrained by a rigid shell. During swimming, the thrust is generated solely by the forelimbs, while the hind limbs are responsible for adjusting posture. An early hypothesis suggested that the sea turtle’s forelimbs functioned like simple paddles, with thrust coming from the drag of water [37]. However, subsequent observational experiments and theoretical analyses revealed that sea turtles employ a lift-based propulsion mechanism, where the thrust is the component of hydrodynamic lift in the forward direction. It is important to note that there is a distinction between the swimming mechanisms of sea turtles and freshwater turtles, with the latter employing a drag-based propulsion method [30].

Observational studies of turtle swimming date back to the 1980s. Davenport et al. compared the swimming movements of green sea turtles and freshwater turtles [30]. They filmed the swimming motions of the turtles in a narrow tank using television cameras and used acetate sheets to plot the postures during swimming. They obtained complete motion trajectories of the forelimb tips during routine swimming and vigorous swimming. The study indicated that green sea turtles tend to have a slight head dip and tail lift while swimming. The forelimbs (referred to as pectoral fins) actively flap to generate thrust, while the hind limbs (referred to as pelvic flippers) trail behind, acting as both a rudder and an elevator, but not contributing to thrust generation. In addition to longitudinal flapping movements, the forelimbs also exhibit lateral rowing movements. During the downstroke, the forelimbs move backward, while during the upstroke, they move forward. The twisting motion of the forelimbs continuously occurs during the flapping process, with more twisting at the distal end than at the proximal end. The downstroke speed is slightly faster than the upstroke speed. By decomposing the flow of water acting on the turtle’s forelimbs during swimming, it was determined that the thrust during turtle swimming originates from hydrodynamic lift rather than drag. Both the upstroke and downstroke of the forelimbs generate thrust, as the sea turtle effectively controls the incoming flow direction through the feathering movement of its forelimbs. Additionally, Davenport’s study included some morphological observations of green sea turtles, such as the forelimbs being about twice the surface area of the hind limbs, being longer, smoother, and flatter. The turtle’s shell is smooth and has hydrofoil-like properties, functioning as a lifting body.

Chun et al. studied the structure and movement process of the sea turtle’s hydrofoil [38]. Through observational experiments, they recorded the motion of the sea turtle’s forelimbs during swimming over a period. The motion cycle of the sea turtle’s forelimbs was divided into four phases: the pronation phase, downstroke phase, supination phase, and upstroke phase, as shown in Figure 2a. The study concluded that the downstroke and upstroke phases are the primary thrust-generating phases during swimming, while the pronation and supination phases serve as transitional phases between the upstroke and downstroke, minimizing resistance and maintaining continuity in the hydrofoil motion. Based on observational data, they decomposed the sea turtle’s forelimb movement during swimming into two independent movements, namely the stroke spin and azimuth spin. These two degrees of freedom (DOFs) are key to all hydrofoil propulsion modes, with the stroke spin generating the primary thrust, while the azimuth spin adjusts the AOA to enhance thrust performance.

Rivera et al. quantified the forelimb motion patterns of loggerhead turtles during flapping styles of swimming and compared them with the rowing styles of swimming patterns of red-eared sliders [40]. They defined and measured the motion angles of the turtle’s forelimbs across different DOFs, distinguishing between the two swimming methods—flapping propulsion used by sea turtles and rowing propulsion used by freshwater turtles. The study found that flapping propulsion, compared to paddle propulsion, has a dominant dorsoventral motion of the forelimbs and exhibits a more prominent feathering motion, confirming that hydrofoil propulsion is a lift-based propulsion method rather than one based on drag. Van der Geest et al. conducted research that refined and supplemented the periodic motion of sea turtle forelimbs [39]. Using a remotely operated underwater vehicle (ROV), they filmed naturally swimming sea turtles in the ocean, distinguishing them from turtles in captivity. They marked the relative positions of the wing tips and rigid shell during the movement cycle and used a simplified physical model to reconstruct the three-dimensional movement of the forelimbs based on the sea turtle’s skeletal features. They divided the forelimb motion of sea turtles into five stages: downstroke (DS), sweep stroke (SS), recovery stroke one (RS1), upstroke (US), and recovery stroke two (RS2), as shown in Figure 2b. Compared to the previously proposed four-stage motion cycle, this study emphasized the dominant horizontal sweep phase in the sea turtle’s forelimb motion. Notably, the results of the three-dimensional motion reconstruction did not fully align with the previously reported two-dimensional kinematic conclusions. The authors noted that they did not observe the typical “8”-shaped motion trajectory in the sagittal plane (Figure 2a), but instead, the “8”-shaped motion trajectory was observed in both the coronal and transverse planes. The “8”-shaped motion trajectory in the sagittal plane has been widely used as a reference for the locomotion pattern of biomimetic sea turtle robots [22,27,41,42,43].

### 2.2. Hydrodynamics of Penguin Wings

Penguins are birds that excel in swimming. They have undergone long-term evolution to fully adapt to underwater locomotion. Penguin wings are adapted for swimming and are no longer suitable for flight, with significant structural differences from the wings of flying birds. Penguin wings feature a thick aerofoil structure supported by the bones of the forearm and hand [44], and this unique structure is the foundation for generating sufficient hydrodynamic lift. The shoulder ball-and-socket joint of penguins allows for various movements, such as flapping, feathering, and sweeping, with the coupling of these movements being key to improving propulsion efficiency.

Observational experiments on the movement of penguin wings during swimming have provided a basis for understanding the swimming kinematics of penguins. Clark and Bemis observed the swimming process of penguins moving forward in a long straight water tank [45]. They recorded the motion trajectory of the wing tips during a complete wingbeat cycle and discussed the relationship between swimming speed and wingbeat frequency, concluding that the two are positively correlated. Hui conducted similar observational experiments, reporting that the penguin wings generate lift during both the upstroke and downstroke, achieved through appropriate control of the AOA [46]. Due to limitations in observational equipment and data analysis methods, early two-dimensional observational results could only provide qualitative analysis of penguin wing kinematics.

Researchers from Tokyo Institute of Technology used multiple underwater cameras and three-dimensional motion analysis software to quantitatively characterize the three-dimensional kinematics of penguin wing motion during free swimming, which further clarified the penguin swimming propulsion mechanism [31,47]. In their observations, they marked points on the penguin’s body and wings (such as the wing base, leading edge, and trailing edge) and observed the penguin swimming freely in an aquarium tank, as shown in Figure 3. Three-dimensional motion analysis revealed that penguin wing motion can be decomposed into three DOFs: flapping, feathering, and sweeping. The movements of each DOF follow a sinusoidal pattern, with the feathering phase occurring 90 degrees ahead of the flapping phase. The sweeping angle remains positive, and its phase is opposite to the flapping phase (i.e., sweeping backward during the downstroke). Additionally, the study measured the bending deformation of the penguin wings during swimming. The bending angle referred to the angle between the inner and outer wing planes. Anatomical studies have shown that there are movable joints in the middle portion of the penguin wings between these two wing planes, similar to the human wrist [48], which allows for bending motion. The results showed that the bending of penguin wings continues throughout swimming, with greater bending during the upstroke than during the downstroke. This bending motion alters the AOA of the wings, further enhancing thrust performance. The study reported that the propulsion efficiency of a bent-wing model was 1.8 times that of a flat-wing model without bending.

Subsequently, Harada et al. used a similar approach to study the turning motion mechanism of penguins [49]. Their research focused on the kinematics of penguin wings during turns and the asymmetry in the motion of the wings on each side. During a turn, the penguin’s body actively tilts outward (with the belly facing the turning side) to generate the centripetal force required for the turn, which contrasts with flying birds. In flight, birds use the lift generated by the upstroke of their wings as centripetal force, causing their backs to tilt toward the turning direction (with their bellies facing outward). This difference arises due to the disparity in the densities of the two mediums, with water being 800 times denser than air. Penguins need to overcome buoyancy through active wing movements [50]. For a turning wingbeat, the upstroke generates lateral force, while the downstroke generates forward force. Compared to forward (straight) swimming, the amplitude of flapping motion increases during turning, while the amplitude of feathering decreases. Considering the asymmetry in the motion of the wings on each side, the inner wingbeat angle is larger than the outer wingbeat angle, producing both yaw and pitch torques. In addition to the wing motion, the penguin’s neck and tail bend inward, acting as rudders to generate yaw torque during turns.

### 2.3. Decoupling Hydrofoil Motion: A Path to Independent DOF Control

Biomimetic robots based on hydrofoil propulsion attempt to replicate the combined movements of hydrofoils to achieve efficient swimming. The combined movement of hydrofoils is a coupling of multiple rotational DOFs. To accurately replicate the locomotion pattern of biological organisms and achieve effective motion control, it is necessary to decouple the combined motion of the hydrofoil into independent movements of different DOFs.

One important method for decoupling the hydrofoil motion is the wing tip (or flipper tip) trajectory method [38,39,51]. This method is based on motion observation, where the swimming process of the biological organism is filmed, and the trajectory of the hydrofoil cross-section during a movement cycle is recorded. The inverse kinematics of the hydrofoil could be solved through relative positions between the body and the wing tip. This process ultimately derives the possible movements of the hydrofoil joints. The advantage of this method is that the process of obtaining the hydrofoil cross-sectional trajectory is relatively intuitive. However, due to the non-uniqueness in the inverse solution, early studies could not fully define the actual motion of the hydrofoil solely based on the wing tip trajectory. Additionally, anatomical studies confirm that the biological structure of the hydrofoil is a coupling of rigid and flexible parts. Sea turtles and penguins have movable joints along the chord-wise direction of their forelimbs. The rear edge portions are unsupported by bones, causing the hydrofoil to bend and twist actively or passively during movement. The deformation of the hydrofoil introduces additional DOFs into the system, further increasing the uncertainty in the actual motion of the hydrofoil. Recently, Van der Geest et al. combined the wing tip trajectory with the structural characteristics of sea turtle forelimbs to perform a three-dimensional reconstruction of their motion, ultimately achieving the decoupling of the sea turtle’s locomotion pattern [39].

In addition to the wing tip trajectory method, three-dimensional motion measurement [31,40,49] can be used to obtain the actual motion of characteristic points or planes of the subject. The specific measurement method is as follows: several characteristic points are selected on the torso and wings of the subject to form characteristic planes, which represent certain parts of the hydrofoil. Cameras, calibrated in advance, are used to record the motion of these characteristic points without affecting the subject. The motion of the characteristic points is reconstructed in three dimensions, and the three-dimensional motion results are then projected onto a two-dimensional plane to obtain the movement angles of the characteristic points in various two-dimensional planes. The results from three-dimensional motion measurement directly reflect the actual motion state of a particular characteristic plane on the subject, providing a more comprehensive definition of the macroscopic movement of part of the hydrofoil. Compared to the wing tip trajectory method, the process of three-dimensional motion measurement is more complex, but it is useful for studying the bending and twisting of the hydrofoil itself.

### 2.4. Locomotion Patterns in Hydrofoils

Figure 4 shows the locomotion patterns of sea turtle and penguin forelimbs. The sea turtle’s locomotion pattern is based on a report by Van der Geest et al. [52], where the curve was derived by approximating the three-dimensional reconstruction of the entire forelimb from the wing tip trajectory. The vertical axis in the figure represents the motion angle of the hydrofoil, while the horizontal axis represents the actual motion time of two wingbeats, with a wingbeat frequency of 0.23 Hz. The penguin’s locomotion pattern is based on a report by Harada et al. [31], where the curve was derived from three-dimensional motion measurements at the base of the hydrofoil. The vertical axis represents the motion angle of the hydrofoil, and the horizontal axis represents the normalized time of one wingbeat of the penguin, meaning that this image does not correspond to the actual wingbeat frequency. The corresponding flap motion of the hydrofoil corresponds to the flap angle, the sweep motion corresponds to the sweep angle, and the feather motion corresponds to the feather angle.

From the relative magnitude of the motion amplitudes, the longitudinal flapping motion dominates, followed by the feathering motion, and the sweeping motion has the smallest amplitude. It should be noted that due to differences in the hydrofoil structures of sea turtles and penguins, the differences in their locomotion patterns are objective. In sea turtles, the flapping motion and sweeping motion of the forelimb occur at different joints, with large-amplitude sweeping occurring at the elbow. For penguins, the longitudinal flapping motion and horizontal sweeping motion occur simultaneously at the shoulder ball-and-socket joint, forming an oblique flapping relative to the body. Three-dimensional motion measurement results show that the trajectory of the penguin’s flapping motion is approximately 75 degrees relative to the horizontal plane. This motion can be decomposed into vertical longitudinal flapping and horizontal sweeping. This coupled motion relationship can be represented by the tilting angle of the flapping motion in the sagittal plane. In the report by Van der Geest et al., the motion angles of each DOF represent the actual motion angle of the hydrofoil relative to the driving part, with the position of zero amplitude corresponding to the natural initial position of the joint. In the report by Harada et al., the motion angles of each DOF are based on a predefined coordinate system before three-dimensional motion measurement. The sweep angle is defined as the angle between the projection of the hydrofoil’s leading edge onto the horizontal plane passing through the root characteristic point and the coronal plane passing through the root characteristic point. In Figure 4b, the continuously positive sweeping angle indicates that the hydrofoil maintains a rearward sweep during motion, with the leading edge always positioned behind the coronal plane passing through the root characteristic point. The feathering motion is expressed as the rotation of the aerofoil part around the axis of the wing. Whether in sea turtles or penguins, the feathering motion is asymmetric in amplitude, with the supination during the upstroke being smaller than the pronation during the downstroke, and this result correlates with earlier observations [30]. The amplitude of feathering in sea turtles is higher than in penguins, and during extreme positions in the motion, it remains for a period of time, providing a smooth transition between the upstroke and downstroke.

In conclusion, as anticipated, the forelimbs of both sea turtles and penguins are highly similar in their locomotion patterns and can be considered to have three independent rotational DOFs. The decoupled locomotion patterns of the biological organisms provide the amplitude of each DOF’s motion and the phase relationships between the DOFs, offering important references for the design of driving mechanisms for biomimetic robots.

### 2.5. A Comparison Between the Hydrofoils

It is common to refer to the sea turtles’ forelimbs and penguin wings as hydrofoils and categorize their swimming methods as hydrofoil swimming, distinguishing them from swimming methods based on oscillation or undulation propulsion. However, there are notable physiological and kinematic differences between the forelimbs of sea turtles and penguins.

First, the forelimbs of turtles exhibit a clear folding angle between the humerus and forearm, similar to the human arm in a bent position [27], as shown in Figure 5b. The aerofoil portion that generates lift is located in the forearm, from the elbow joint to the flipper tip. Regarding the locations of different DOFs, the longitudinal flapping movement occurs at the shoulder joint, the horizontal sweeping movement occurs at the elbow joint, and the axial feathering movement is expressed through torsion in the forearm. In contrast, during swimming, penguin wings are almost fully extended, with the aerofoil portion extending from the wing base to the wing tip [31]. All rotational DOF movements occur at the shoulder joint [48], as shown in Figure 5a.

Secondly, there are significant differences in their posture control methods. The rigid shell of sea turtles restricts body movement, and they primarily rely on the hind limbs for posture control (the head and tail may also help). In contrast, penguins actively engage their head, spine, and feet in posture control, which allows them to perform more agile swimming movements, such as quick turns and leaping out of the water [49]. Additionally, the ways in which penguins and sea turtles overcome buoyancy differ, leading to differences in how their hydrofoil movements contribute to propulsion. Penguins inhale sufficient air during dives to protect their lungs and store oxygen. According to estimates by Ponganis et al., when the volume of air in a penguin’s lungs reaches the limit of the air sacs, the penguin’s density is approximately 0.7 g/mL, lower than the density of water, which means penguins cannot achieve negative or neutral buoyancy underwater by adjusting their body density [54]. As a result, penguins must generate downward force through a powerful upstroke to overcome their buoyancy [50]. Among all diving animals that rely on wing propulsion (such as penguins, alcids, and sea turtles), penguins are the only animals in which the upstroke contributes equally to or even more than the downstroke in generating propulsion [31]. Sea turtles, however, can effectively adjust their buoyancy by controlling the air content in their lungs, negating the need for an upstroke to overcome positive buoyancy. For sea turtles, thrust generation from hydrofoil motion primarily occurs during the downstroke, with the downstroke contributing more to propulsion than the upstroke.

Lastly, regarding the wingbeat frequency and swimming speed of sea turtles and penguins, both species show objective differences between different individuals. Overall, penguins have a wingbeat frequency about twice that of sea turtles, and their swimming speed is typically two to three times faster. According to recent observational data, the wingbeat frequency of a naturally swimming green sea turtle is about 0.23 Hz, corresponding to a swimming speed of approximately 0.6 m/s [39]. Penguins, on the other hand, have a much higher wingbeat frequency, typically ranging between 1.43 and 2.50 Hz, with average swimming speeds greater than 1 m/s. Harada et al. emphasized that this record represents penguins’ routine swimming speeds, with speeds in extreme environments (such as foraging or escaping predators) being higher than those recorded in the study (ranging from 0.83 m/s to 2.07 m/s) [31]. The speed of an emperor penguin while hunting is about 2.3 m/s [55].

In summary, upon reviewing the swimming kinematics of sea turtles and penguins, both species retain their distinct movement characteristics while sharing similarities. From the perspective of overall swimming method and thrust generation mechanism, both penguins and sea turtles rely solely on the movement of their forelimbs to generate lift-based thrust. Their forelimb movement modes are similar, with longitudinal flapping motion dominating and the ability to actively change the AOA through feathering motion to enhance thrust performance. Overall, hydrofoil propulsion, as a continuous propulsion mode, offers advantages in propulsion efficiency. During hydrofoil motion, the movement of the forelimbs along the dorsal–ventral axis generates significant lateral forces. During forward swimming, the symmetric movement of the hydrofoils on both sides cancels out the lateral forces, emphasizing the forward thrust. In fact, the asymmetric movement of the hydrofoils on both sides can generate substantial yaw torques, which provides the possibility for agile swimming. Finally, based on observations of natural organisms, both sea turtles and penguins can swim long distances underwater while consuming relatively little food. The low energy consumption observed during swimming is speculated to be related to their unique propulsion methods.

## 3. Biomimetic Hydrofoil Driving Mechanism

This chapter summarizes the implementation of various driving mechanisms based on hydrofoil propulsion. The ideal driving mechanism should both enable necessary movements of the hydrofoil and generate sufficient propulsion thrust. Generally, when designing the driving mechanism, several factors need to be considered, including the system’s sealing, compactness, and the motion range of each DOF. Sealing refers to the waterproof performance of the entire system. For underwater vehicles, traditional propeller-based thrusters have relatively mature engineering solutions for sealing. However, for hydrofoil-based propulsion systems using various driving mechanisms, the sealing design needs to be carefully considered in relation to the specific mechanism structures and application scenarios. Compactness requires the electromechanical system to have a high space utilization rate, and the design should avoid bulky structures as much as possible to meet the overall specifications of the system. Finally, the motion range of the mechanism refers to the amplitude of each DOF, which impacts the propulsion efficiency of the biomimetic robot. The dimensionless Strouhal number can be used to indirectly characterize the propulsion efficiency of marine animals. A common expression for the Strouhal number is St=Lf/v, where St represents the Strouhal number. For underwater flapping wings, *L* is the characteristic length (usually represented by the amplitude of motion in units of length), *f* is the flapping frequency, and *v* is the relative flow velocity of the fluid to the wing. Existing research shows that for most marine organisms, the Strouhal number ranges from 0.2 to 0.4, which corresponds to high propulsion efficiency [56]. Similarly, for biomimetic underwater robots, the amplitude of motion affects the characteristic length, thus influencing the propulsion efficiency. When designing the driving mechanism, it is important to ensure that the mechanism can achieve sufficient motion amplitude and flapping frequency, so that the Strouhal number falls within the range associated with high propulsion efficiency.

### 3.1. Sea Turtle-Inspired Driving Mechanism

For multi-input systems that require the coordination of multiple motors, direct motor drive is always a solution worth considering. This type of mechanism uses a designed rigid framework to connect the various motors, with the motors directly driving the motion of each DOF. The system does not involve any additional transmission components. This design reduces the space occupied by transmission parts, making the structure relatively compact and easier to implement. Many studies have attempted to apply this solution, such as Baines et al.’s amphibious robot turtle (ART) [32] and Li et al.’s biomimetic sea turtle [57], as shown in Figure 6a,b. The drawback of this structure is that the stacked motors can lead to a bulky connection between the hydrofoil and the body, which increases the requirements for sealing design. Typically, flexible rubber bellows or silicone gel sleeves are used to enclose the entire driving frame for sealing. However, this sealing solution can somewhat affect the streamlined structure of the overall system.

When designing the biomimetic sea turtle hydrofoil propulsion system, Font et al. summarized four different driving mechanisms and compared their potential: the four-bar mechanism, the ball-and-socket mechanism, the differential gear mechanism, and the translational pulley mechanism [41]. The design was based on earlier research that provided two-dimensional kinematic observations of the sea turtle’s wing tip trajectory, specifically the “8”-shaped motion trajectory in the sagittal plane. Specifically, the amplitude of the four-bar mechanism’s motion depends on the length relationships of the rods, making it difficult to control the motion amplitude flexibly. The challenge with the ball-and-socket mechanism is the complexity of generating the motion sequence. The differential gear mechanism requires an additional motor to control the horizontal movement, taking up considerable internal space and being difficult to seal. The translational pulley mechanism has poor stability, as rope wear can lead to uncertainty in the movement of hydrofoils. After considering amplitude, compactness, and motor torque, Font et al. chose the ball-and-socket mechanism, as shown in Figure 6c. This mechanism connects three motors through a nested rigid frame, driving the motion of each DOF. Specifically, M1 controls the horizontal rowing motion, M2 drives the vertical flapping motion, and M3 controls the feathering motion around the axis. The authors reported that this mechanism is highly compact, provides sufficient motion amplitude, and facilitates effective sealing.

Wang et al. also employed a similar ball-and-socket structure in their study on the asymmetry of hydrofoil feathering motion, as shown in Figure 6d [58]. The design was based on the locomotion pattern of sea turtle forelimbs proposed by Van der Geest et al. [52]. The decoupled locomotion pattern simplified the generation of motor motion sequences, eliminating the need for motion simulations to produce specific trajectories, thus addressing the challenges of ball-and-socket mechanism applications mentioned by Font et al. The realization of the feathering motion is similar to Font et al.’s design, being directly driven by a motor connected to the hydrofoil. The difference lies in the driving mechanism for the flapping motion, where a set of mutually perpendicular bevel gears changes the transmission direction of the flapping motion by 90°, further reducing the front-to-back distance.

Rope-driven mechanisms have been widely studied due to their lightweight nature and potential applications in structures with rigid–flexible coupling. Figure 6e depicts a four-DOF robotic sea turtle forelimb designed by Yan et al. [59]. Three of the DOFs are used to achieve the overall motion of the hydrofoil: A yaw servo motor located at the shoulder drives the rowing motion. A roll servo motor and a pitch servo motor located at the elbow drive the feathering and flapping motions, respectively. The robotic forelimb is designed with variable stiffness, independently controlled by an additional motor. All the servo motors’ power transmission is driven by the output shafts’ spools. This rope-driven structure offers high space utilization but places high demands on the motor’s output torque and the durability of the rope material.

In addition to motor-driven mechanisms, flexible deformable hydrofoils powered by smart soft composite (SSC) materials are another potential solution. This approach emphasizes the flexible characteristics of real hydrofoils, using the deformation movement of the hydrofoil itself to replace the limb movement generated by a driving mechanism. Figure 6f depicts a flexible deformable hydrofoil designed by Song et al. [60], based on two swimming gaits of the green sea turtle observed by Davenport [30]. The SSC actuator is composed of a drive shaft (SMA), a support structure (ABS), and a flexible polymer (PDMS). Based on laminate theory, the stiffness of the support structure can be controlled by adjusting the stacking angle of the support structure, thereby influencing the direction of deformation. By controlling the current input to the SMA, the hydrofoil can deform to achieve the desired movement gait. The movement of the SSC-driven flexible hydrofoil is smooth and continuous. This drive method surpasses traditional motor drives in terms of lightweight design and space utilization, but it lacks the flexibility and precision in controlling the motion of each DOF as motors do. Furthermore, due to limitations in movement frequency, the thrust generated by hydrofoil deformation is limited, resulting in relatively low swimming speeds.

Van der Geest et al. performed reverse engineering on biological sea turtles and proposed a biomimetic robotic sea turtle [52]. The driving mechanism of this robotic sea turtle has three independently controlled joints and is highly similar to a real sea turtle, as shown in Figure 6g. The flapping and sweeping motions are generated by different joints, with the ratio of the two forelimb segments approximating that of a real sea turtle. The hydrofoils are made of flexible materials and can undergo axial twisting under the drive of motors to achieve feathering motion. The authors reported that the flipper tip trajectory generated by this driving mechanism closely matches the natural flipper tip trajectory of a sea turtle during swimming.

### 3.2. Penguin-Inspired Driving Mechanism

Sudki et al. designed a three-DOF spherical joint to simulate the movement of a penguin’s shoulder joint (Figure 7a) [61]. This mechanism is similar to the motion platform of a parallel robot, capable of driving the end effector to perform arbitrary rotational motion within a 60-degree cone. Compared to serial robots, the fixed actuators (motors) reduce the impact of inertial forces, making it easier to achieve high acceleration and precise control. The drawback is that the motion chain of the parallel platform occupies more internal space, making integration more challenging.

Shen et al. designed a three-DOF robotic penguin wing based on the results of three-dimensional motion measurements of penguin swimming (Figure 7b) [47]. This penguin wing can achieve flapping, feathering, and pitch motions. The flapping and feathering motions are transmitted through a differential gear mechanism, ensuring high compactness. The pitch motion is independently driven by an additional servo motor, which changes the angle of the flapping motion in the sagittal plane, replacing the traditional sweeping motion. In fact, based on the three-dimensional motion measurement results, the phase relationships between the motions of each DOF are determined. The pitch motion locks the phase relationship between the vertical flapping motion and the horizontal sweeping motion, maintaining a phase difference of 0 between the two, ultimately forming an oblique flapping relative to the body. This phase-locked design avoids the horizontal rotational movement of the entire differential gear structure, reducing the difficulty of sealing.

## 4. Biomimetic Flexible Hydrofoils

Bio-inspired flexible flapping foils have emerged as promising alternatives to traditional rigid propellers by emulating marine organisms’ propulsion strategies. Recent advances [62] reveal that controlled flexibility enhances propulsion efficiency through three synergistic mechanisms: (1) passive adaptation to unsteady flows, (2) vortex-induced energy recovery, and (3) resonance-enhanced thrust generation. However, practical implementation faces dual challenges: the strong non-linearities in fluid–structure interaction (FSI) control [63] and material fatigue under high-cycle hydrodynamic loading [64]. This section systematically analyzes the energy conversion mechanisms while addressing these implementation barriers. Representative flexible hydrofoil designs demonstrating practical solutions are subsequently presented.

### 4.1. Efficiency Advantages of Flexibility

The hydrodynamic superiority of flexible hydrofoils stems from three interdependent mechanisms that optimize energy transfer through controlled structural compliance. Chord-wise flexure dynamically modulates vortex evolution by delaying flow separation and amplifying trailing-edge pressure differentials, generating 18–25% higher pressure gradients than rigid foils [62]. Simultaneously, adaptive camber deformation synchronizes effective angles of attack ($\alpha_{\mathrm{eff}}$) with hydrodynamic cycles, maintaining $\alpha_{\mathrm{eff}}$ in the optimal 22°–35° range during 85% of wave periods [65]. This synchronization reduces dynamic stall occurrences by 60% while enabling passive energy recovery, where 18% of bending energy converts into thrust via elastic potential storage [64].

Bio-inspired motion patterns exploit these mechanisms through two distinct operational modes. The penguin-inspired “8”-shaped trajectory enhances operational envelopes by 40% through dynamic Strouhal number matching (St = 0.25–0.35), with particle image velocimetry (PIV) confirming 30% expanded vortex cores and 40% wider high-velocity zones [31,63]. Conversely, the turtle-like semi-activated oscillation mode sustains efficiency across variable flows via optimized kinematics (pitching amplitude: 50°–80°; reduced frequency: 0.15–0.25) [39]. Modified unsteady vortex lattice method (m-UVLM) simulations demonstrate bimodal transitions between high-thrust and high-efficiency regimes, achieving 8% agreement with towing tank measurements [66].

Compared to screw propellers, flexible hydrofoils exhibit context-dependent advantages. At low speeds (<5 m/s), they achieve 33% higher energy efficiency by suppressing turbulent dissipation and harvesting elastic energy [67], while 50% acoustic signature reduction stems from eliminated tip vortex losses [68]. Though limited by material fatigue at high speeds (>10 m/s) [63], their thrust-to-power ratios under unsteady conditions surpass propellers by 1.8× [69]. This performance improvement highlights their potential for systems utilizing wave energy where conventional propellers suffer from flow separation and cavitation.

### 4.2. Adaptive Design Trade-Offs

The development of adaptive hydrofoil propulsion systems faces three primary challenges: control system non-linearities, stringent material requirements, and the delicate balance between passive deformation and active control. Advanced 3D fluid–structure interaction (FSI) simulations reveal that span-wise twist angles (15–22%) enhance lift through passive camber morphing, but simultaneously induce unpredictable vortex shedding frequencies [70]. This dynamic coupling causes control dimensionality explosions in six-degree-of-freedom systems, reducing computational efficiency by 60% despite hybrid zero-order-approximation LQR optimization [71]. Frequency lock-in phenomena further amplify pressure pulsations by 40% when hydrofoil vibrations synchronize with cavity shedding cycles (Δf≥8.7 Hz) [72,73].

The material durability of flexible hydrofoils presents a critical research gap in marine applications. Current implementations on underwater vehicles lack sufficient experimental validation regarding long-term material fatigue and synergistic seawater corrosion effects. This stands in contrast to turbine foil applications, where substantial field data and operational performance metrics have been systematically documented [74]. Numerical investigations have provided preliminary insights into hydrodynamic interactions, with computational models revealing alternating stress amplitudes exceeding 120 MPa at flow velocities of 6 m/s, accompanied by stress concentration factors ranging from 2.3 to 3.8 near critical flexure joints [68]. Emerging material systems face compounded challenges in marine environments: Piezoelectric composites demonstrate 15% functional degradation from chloride ion penetration within 90-day exposure periods [73], while optimized carbon fiber laminates (0°/45°/90° stacking sequence) maintain residual delamination risks under $10^{6}$-cycle saline immersion tests [75]. Particularly concerning for flexible hydrofoil dynamics, accelerated corrosion–fatigue interactions have been shown to reduce ultimate tensile strength by 35% in marine-grade aluminum alloys after 500 h seawater exposure [75]. These findings highlight the critical need for material-specific validation protocols addressing the unique stress–corrosion coupling inherent in flexible hydrofoil flapping mechanisms.

Design optimization requires careful trade-off between passive adaptability and active control. Bio-inspired arc textures reduce cavitation volumes by 40% through vortex stabilization, albeit at the cost of 12–18% lift reduction [72]. Semi-active architectures achieve 35% energy savings compared to fully active systems, but require 30% longer stabilization during flow transients. Current solutions employ digital twins to optimize Strouhal numbers (St=fA/U) within 0.25–0.35, combined with real-time CFD-FEA frameworks that reduce prototype costs by 70% [68]. Emerging deep learning models show promise in addressing non-linear FSIs, with modified CNNs achieving >90% accuracy in pressure field predictions [76]. Nevertheless, persistent limitations in seawater durability and control bandwidth (0.5–3 Hz) highlight the need for integrated solutions combining physics-informed control, corrosion-resistant materials, and adaptive curvature modulation.

The pursuit of optimal hydrofoil performance necessitates careful balancing between passive morphing capabilities and active control complexity. Three representative bio-inspired hydrofoil designs exemplify the current state-of-the-art approaches and their inherent compromises:

(1) Humpback Whale-Inspired Hydrofoil [63]: Featuring sinusoidal leading-edge protrusions, as shown in Figure 8a, this design reduces cavitation volume by 40% through controlled vortex stabilization. While achieving exceptional fatigue resistance (>10^6^ cycles at 8 Hz), its wavy geometry incurs 12–18% lift reduction at low angles of attack (α < 8°), necessitating compensatory active pitch control systems. The carbon-fiber-reinforced silicone composite demonstrates excellent corrosion resistance, yet suffers 22% bending stiffness reduction after 500 h seawater exposure due to chloride ion penetration.

(2) Flax Fiber Composite Hydrofoil [77]: Utilizing topology-optimized “double-I” beam structures with >50% bio-composite content, this sustainable design achieves 34% mass reduction while maintaining comparable strength-to-weight ratios. Although the concurrent XFOIL-CFD framework reduces prototyping costs by 70%, the design neglects critical seawater degradation effects—experimental data reveal 15% moisture-induced dimensional instability and delamination risks at flow speeds >8 m/s.

(3) Sea Turtle-Inspired Morphing Hydrofoil [78]: Mimicking chelonian foreflipper kinematics, as shown in Figure 8b, this design employs graded-stiffness polyurethane layers for passive curvature modulation, achieving 85% wave-energy conversion efficiency in 1–2 m waves. While reducing mechanical complexity by 40% compared to SPFF systems, its viscoelastic core material exhibits 30% hysteresis loss during rapid deformation cycles (>2 Hz), and maximum thrust (120 N) proves insufficient for vessels exceeding 5 tonnes.

These implementations collectively demonstrate that current bio-inspired solutions achieve specialized performance enhancements at the expense of either material durability (e.g., seawater degradation), operational range (e.g., limited thrust scalability), or energy efficiency (e.g., hysteresis losses). The persistent 18–25% performance gap between laboratory prototypes and biological counterparts underscores the need for hybrid approaches combining advanced materials science with adaptive control algorithms.

**Figure 8 biomimetics-10-00272-f008:**
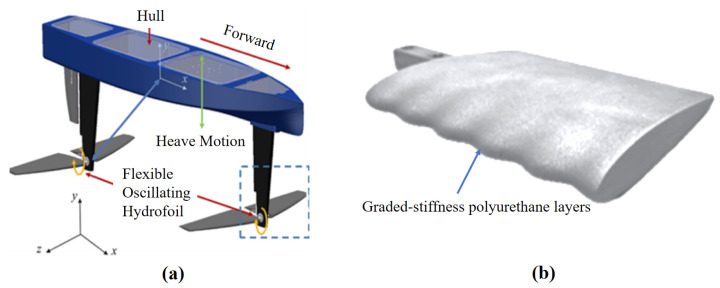
(**a**) Humpback whale-inspired hydrofoil [63]; (**b**) sea turtle-inspired morphing hydrofoil [78].

### 4.3. Prospects for Flexible Hydrofoil Propulsion

The advancement of flexible hydrofoil propulsion systems requires coordinated solutions to three persistent challenges. Control system complexities rooted in non-linear dynamics may benefit from methodologies developed for aerial flapping wings [79] and turbine hydrofoils [80], though their transferability requires validation under aquatic flow conditions [76]. Material durability limitations, particularly strain fatigue accumulation and saltwater corrosion mechanisms, remain understudied due to insufficient long-term (>10,000 h) experimental data [81]. The design trade-off between passive adaptability and active control remains inadequately addressed due to the scarcity of functional prototypes—existing variable-stiffness designs [59] lack integrated actuation, while bio-inspired concepts [52] await hydrodynamic performance verification.

Three synergistic research directions emerge to overcome these barriers. First, hybrid control architectures combining reinforcement learning [82] and adaptive sliding-mode control could address non-linear fluid-structure interactions. Second, accelerated aging protocols simulating combined mechanical–chemically aggressive environments must be developed, building upon recent corrosion-fatigue models [83]. Third, modular hydrofoil platforms enabling real-time stiffness modulation (0.1–10 GPa range) should be prototyped using multi-material additive manufacturing [84]. Concurrently, digital twin frameworks require enhancement through embedded strain sensing arrays to validate passive–active transition thresholds observed in simulations [68].

The path forward necessitates parallel advancement in computational models and physical implementations. While deep learning approaches achieve 92% accuracy in transient load predictions [76], their integration with physics-based FSI solvers remains unexplored. Field trials of bio-mimetic prototypes must quantify actual performance metrics against conventional propellers through standardized hydrodynamic testing. As no established ISO standard currently exists for flapping-wing propulsion systems, the evaluation framework can adopt methodologies from standard for hydraulic performances tests for waterjet propulsion sysytem [85], which provides validated protocols for measuring thrust efficiency, flow dynamics, and cavitation resistance under controlled marine conditions. Through such multidimensional efforts, flexible hydrofoils could ultimately realize their theoretical efficiency advantages while achieving operational reliability in marine environments.

## 5. Hydrofoil-Based Biomimetic Robots

### 5.1. Sea Turtle Robot

Biomimetic hydrofoil propulsion robots typically mimic the swimming patterns of sea turtles or penguins. Over the past period, various underwater biomimetic robots have been developed, including those inspired by sea turtles or penguins. It is important to note that while many sea turtle robots have been realized, not all robots inspired by sea turtles use hydrofoil propulsion for swimming. Some robots have met the characteristics of the biological organism in terms of appearance and limb structure, but the actual locomotion patterns have been overlooked. Due to the previous lack of clarity in the hydrofoil locomotion pattern, there have been few attempts to replicate the motion of biological organisms in robots. According to the locomotion pattern in Figure 4a, the forelimbs of a sea turtle have three rotational DOFs. Currently, only a few robotic sea turtles possess forelimbs with all three DOFs. Most underwater robots inspired by sea turtles use biomimetic hydrofoils with one or two DOFs for propulsion. Since the simple motion of tilting and flapping alone is insufficient to replicate the coupled motion of real hydrofoils, biomimetic hydrofoils with a single DOF cannot reproduce the swimming behavior of a biological sea turtle. Robots equipped with two-DOF hydrofoils can adjust the AOA while flapping. The biomimetic hydrofoil with 2 DOFs covers the main movements of a biological hydrofoil (i.e., vertical flapping and axial feathering), meeting the basic conditions required to replicate the swimming behavior of a real sea turtle. Finally, robots with three-DOF hydrofoils introduce a horizontal sweeping motion in addition to the flapping and feathering motions. These robots exhibit the highest biomimetic potential in attempts to reproduce the excellent propulsion effects of hydrofoil-based movement.

Among all underwater robots that use 3-DOF hydrofoil for propulsion, ETH’s Naro-tartaruga represents a pioneering attempt (Figure 9a) [86]. This robot features a mature modular design, with an internally sealed aluminum tube as its core structure, and a removable outer shell with a streamlined form resembling that of a real sea turtle. Naro-tartaruga uses a pair of sophisticated shoulder joints for propulsion, with each shoulder joint possessing three rotational DOFs. Siegenthaler et al. validated the reliability of this 3-DOF biomimetic fin and were the first to attempt to control an underwater robot driven by fins with 3 DOFs. All the driving joints of Naro-tartaruga are concentrated on the shoulders of the robot. From the biomimetic perspective, compared to sea turtles, the distribution of these driving joints is more “penguin like”.

Yan et al. proposed a robotic sea turtle with variable-stiffness fins (Figure 9b) [59]. The robot follows a modular design, with a center-of-mass adjustment mechanism and a buoyancy adjustment mechanism. The driving mechanism for each fin includes three rotational DOFs, and the joint distribution refers to the skeletal structure of a real sea turtle. The variable-stiffness front fins generate thrust, while the passively deforming flexible rear fins control the direction of movement.

Van der Geest et al.’s soft robotic green sea turtle replicates the movement of a green sea turtle (Figure 9c) [52]. The driving mechanism of this robotic turtle, as introduced in the previous section, is powered by three independent servos. When the hydrofoils flap at a frequency of 0.23 Hz, the robotic turtle swims at a speed of 0.6 m/s, which is highly consistent with the observed natural swimming speed of sea turtle. The corresponding Strouhal number for this flapping frequency and speed is 0.24, which falls within the efficient propulsion range observed for swimming animals.

### 5.2. Penguin Robot

As shown in Figure 4b, compared to the forelimbs of sea turtles, the amplitude of the penguin wing’s sweep motion is much smaller. Most penguin-inspired robots do not consider the horizontal motion of the penguin wings, focusing only on the vertical flapping motion and the rotational feathering motion. Two-DOF flapping wings have been proven to be an effective propulsion device. With sufficient amplitude and frequency, appropriate coupling of the flapping and feathering motions can achieve excellent propulsion and high maneuverability.

The AquaPenguin developed by FESTO is groundbreaking (Figure 10a) [87]. The AquaPenguin has a flexible torso that can move in any direction and uses a pair of biomimetic wings to generate thrust. The flapping motions of the wings are controlled by a single motor synchronously, while the feathering motion of each side can be controlled independently to achieve maneuverability. Additionally, the AquaPenguin is equipped with pressure sensors and sonar for environmental perception and autonomous navigation.

Pan et al. developed a penguin-inspired swimming robot, as shown in Figure 10b [88]. Each wing of the robot has two DOFs and can independently control the flapping and feathering motions. The robot includes a buoyancy adjustment system and an air lubrication system. The air lubrication system, based on compressed cylinders, is located at the middle and rear parts of the robot body. The air lubrication system can generate bubbles around the body of the robot to reduce surface friction. The authors reported that the designed air lubrication system effectively increases the robot’s speed.

Shen et al. optimized the structure of the 3-DOF robotic penguin wing proposed in previous research, merging the pitch control of the two wings [81]. Each wing has two DOFs, and the pitch angle of the flapping plane is controlled simultaneously by an additional motor for both sides, as shown in Figure 10c. The shoulder joint, based on a differential gear mechanism, covers the actual motion range observed in the three-dimensional motion analysis of real penguins. The entire driving mechanism is housed inside the robot, with rubber bellows and hot-melt adhesive used for flexible sealing at the connection between the wings and the body.

Shimooka et al. developed an agile robotic penguin, as shown in Figure 10d [89]. The robot’s driving mechanism is similar to their previous design [47], with each wing having two DOFs, allowing independent control of both the flapping and feathering motions. Instead of using commercial non-waterproof servos, specially designed submersible geared servomotors were employed. The driving mechanism is exposed to water, offering a greater range of motion. The flapping amplitude of the wings can reach 180°, while the feathering motion allows for unrestricted 360° rotation. By applying different movement strategies, the robotic penguin is capable of executing fast rolling, pitching, yawing, and braking, demonstrating high maneuverability.

In conclusion, biomimetic robots inspired by sea turtles and penguins have demonstrated significant advancements in replicating the complex locomotion patterns of these animals. The development of hydrofoil-based propulsion systems has proven crucial for achieving efficient movement, with varying DOFs in the fins or wings directly influencing the robots’ propulsion performance. While early attempts often overlooked the coupled motions of real biological organisms, recent designs incorporating two or three DOFs in hydrofoils have shown promising results in mimicking the swimming behaviors of sea turtles and penguins. Furthermore, the integration of advanced technologies, such as flexible actuators and modular designs, has enhanced the versatility and maneuverability of these robots, highlighting their potential for a range of underwater applications. Despite these advancements, further refinements in propulsion mechanisms and biomimetic accuracy remain necessary to fully replicate the natural efficiency and agility observed in living organisms. All the robots discussed are summarized in Table 1, “N/A” in the table denotes data that are not yet publicly available.

Table 1 presents the swimming speeds of several implemented prototypes, with some advanced robotic systems achieving speeds comparable to their biological counterparts. In terms of swimming efficiency, the performance of biological organisms can be indirectly evaluated using the Strouhal number. For a given species, there is an inherent variation in Strouhal numbers across individuals and swimming behaviors. According to recent observational studies, the average Strouhal number for penguins during straight-line swimming is approximately 0.26, while that of sea turtles is around 0.24, indicating similar efficiency levels. For biomimetic robots, swimming efficiency can theoretically be assessed based on energy consumption. However, since most existing prototypes are still in the proof-of-concept stage, such data remain largely unavailable. Furthermore, differences in power calculation methods and drag measurement techniques across studies hinder meaningful comparisons of efficiency metrics.

## 6. Conclusions

This paper focuses on the swimming mechanisms of sea turtles and penguins, summarizing the current research on hydrofoil-propelled robots. It begins with a review of the swimming kinematics of both sea turtles and penguins, where the superior swimming performance exhibited by these biological organism serves as the primary motivation and foundation for biomimetic research. The study also reviews the methods for analyzing hydrofoil locomotion patterns and summarizes the decoupled motion patterns of hydrofoils in sea turtles and penguins. Decoupling the combined motion of the hydrofoil can effectively guide the design of driving mechanisms. A review of the driving mechanisms reveals that current hydrofoil drive systems can achieve coupled motions with multiple DOFs. Finally, the paper reviews several existing hydrofoil robots. The biomimetic forelimbs of sea turtle robots typically possess three DOFs, while the biomimetic penguin wings in penguin robots usually have two DOFs. The design of these robots shows a strong correlation with the decoupled motion patterns of the biological organism.

Sea turtles and penguins are both excellent swimmers. Researchers have developed hydrofoil-propelled robots in an attempt to replicate their swimming patterns. For a long time, wing tip trajectories have been a significant area of research in both biology and biomimetics. These observations are crucial for understanding the swimming mechanisms of organisms. However, in the field of biomimetic mechanism design, the results of motion observations provide only qualitative conceptual guidance and fail to offer quantitative data that can serve as a foundation for mechanism design.

A review of the current state of research in hydrofoil kinematics reveals that recent efforts have achieved key results. Researchers have combined motion observations of biological organisms with their skeletal features, applying additional constraints to the inverse kinematics of wing tip motion. In other words, biomimetic hydrofoils in motion should adhere to the anatomical principles of the biological organisms. Additionally, some studies have attempted to quantitatively measure the 3D motion of organisms, gathering data from the root of the hydrofoil to avoid the additional biases introduced by solving inverse kinematics. These studies have introduced new perspectives on understanding the swimming patterns of natural organisms, and the decoupling of hydrofoil motion provides effective guidance for the independent DOFs control of hydrofoils.

The decoupled locomotion patterns suggest that biomimetic hydrofoils should possess three DOFs to achieve vertical, horizontal, and rotational movement. While 2-DOF hydrofoils have been widely studied, a review of existing 3-DOF driving mechanisms reveals common design principles. Direct drive and ball-and-socket joints are widely adopted methods, while rope-driven and flexible drives are also feasible approaches. It is important to note that adding additional DOFs to the driving mechanism can enhance its biomimetic potential, allowing for more natural swimming postures. However, this also increases the complexity of the robot’s sealing design. For underwater robots, waterproofing is a critical engineering issue. Mechanical seals, flexible seals, and waterproof servos are the primary directions of exploration.

Oscillating propulsion-based underwater biomimetic robots tend to reach extreme performance. BCF-mode robots generate thrust through high-frequency oscillations of their bodies or tails, achieving high swimming speeds and excellent acceleration. However, as yaw control and forward thrust are coupled in BCF-mode robots, achieving precise attitude control presents significant challenges. In contrast, MPF-mode robots independently control the biomimetic fins on both sides of their bodies to decouple thrust, providing more stable attitude control. However, the lower oscillation frequency limits their swimming speed. The rich locomotion patterns of hydrofoils offer hydrofoil-propelled organisms the potential to balance high-speed movement with precise control, enabling agile swimming. The design of hydrofoil-propelled biomimetic robots reviewed in this paper shows a strong correlation with the decoupled motion patterns of biological organisms. These robots hold the potential to achieve breakthroughs in swimming performance.

## Figures and Tables

**Figure 1 biomimetics-10-00272-f001:**
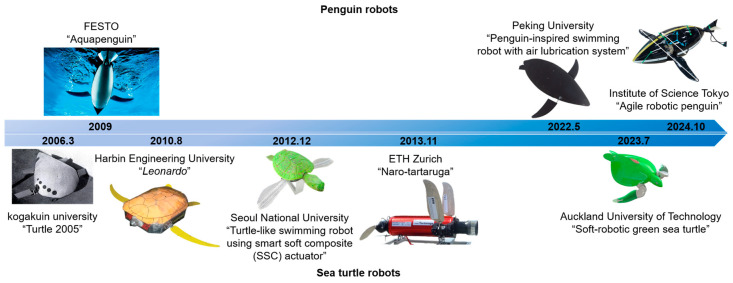
The developmental histories of hydrofoil robots.

**Figure 2 biomimetics-10-00272-f002:**
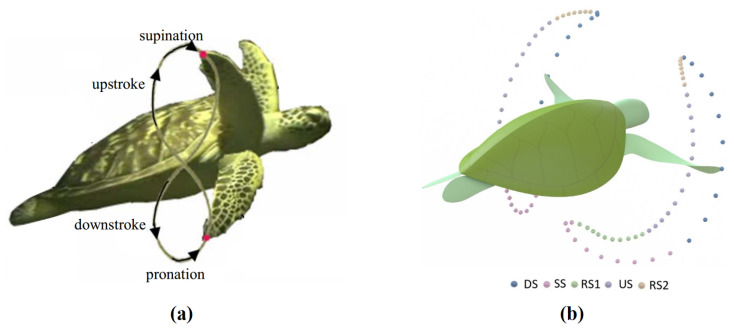
The motion cycle of a sea turtle’s hydrofoil. (**a**) The four-phase cycle proposed by Chu et al. and the “8”-shaped wing tip trajectory in the sagittal plane. The four phases are the pronation phase, downstroke phase, supination phase, and upstroke phase [38]. (**b**) The five-phase cycle observed by Van der Geest et al., consisting of downstroke (DS), sweep stroke (SS), recovery stroke one (RS1), upstroke (US), and recovery stroke two (RS2) [39].

**Figure 3 biomimetics-10-00272-f003:**
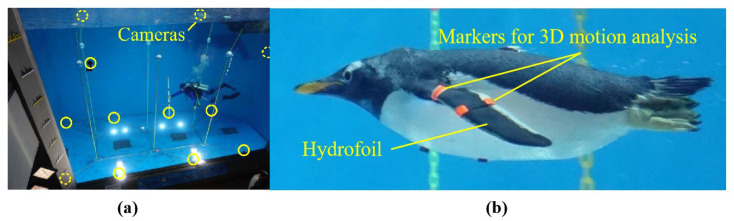
(**a**) Field of the 3D motion measurement in the aquarium, the circled parts are cameras. (**b**) Swimming penguin with 3D motion analysis [47].

**Figure 4 biomimetics-10-00272-f004:**
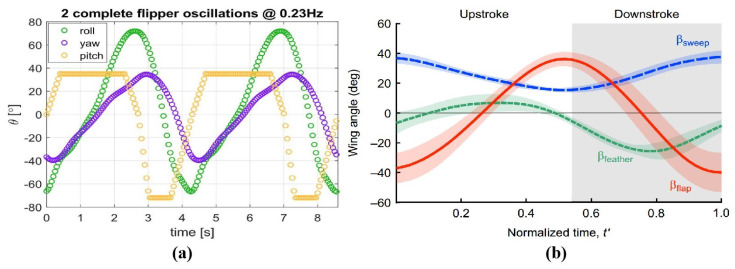
The locomotion patterns of sea turtle and penguin forelimbs. (**a**) The locomotion patterns of sea turtle forelimbs at a frequency of 0.23 Hz [52]. (**b**) The locomotion patterns of penguin wings in one wingbeat [31].

**Figure 5 biomimetics-10-00272-f005:**
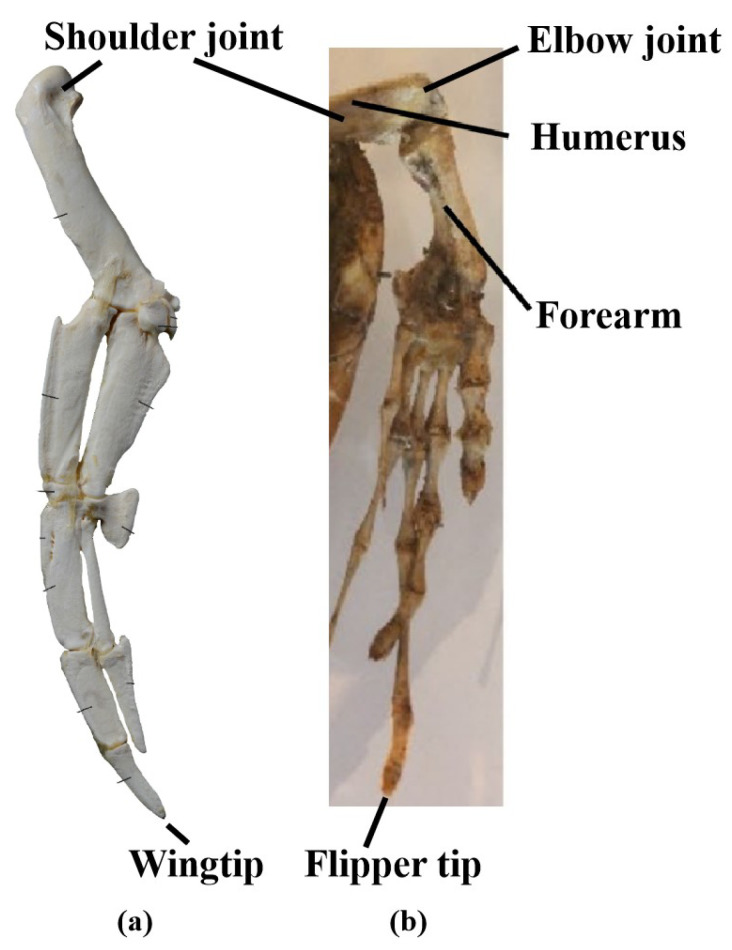
(**a**) Humboldt penguin wing skeleton [53]. (**b**) Hawksbill sea turtle flipper skeleton [39].

**Figure 6 biomimetics-10-00272-f006:**
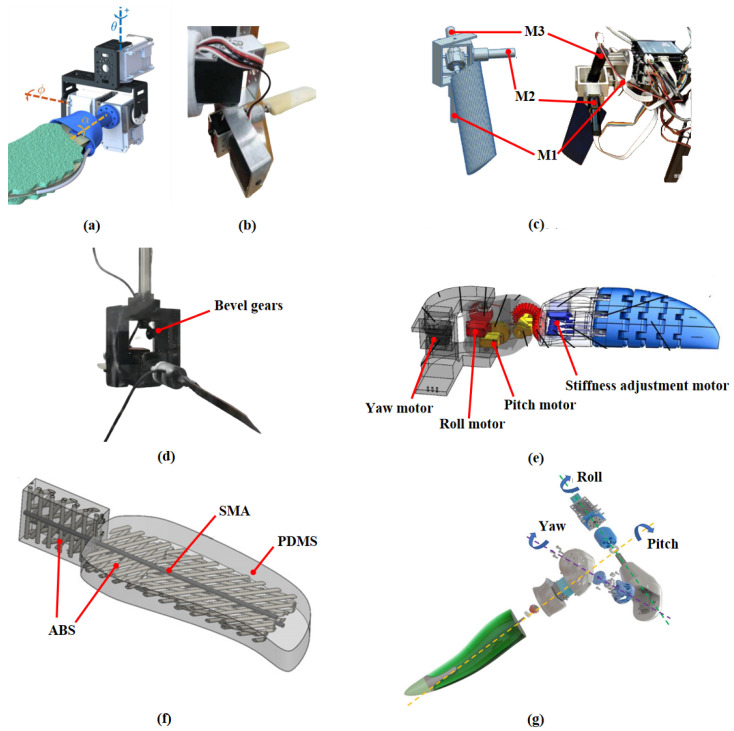
Sea turtle-inspired driving mechanisms: (**a**) The direct driving mechanism of the ART [32]. (**b**) The direct driving mechanism of Li et al’s robotic sea turtle [57]. (**c**) Ball-and-socket mechanism by Font et al. [41]. (**d**) Ball-and-socket mechanism by Wang et al. [58]. (**e**) The rope-driven mechanism [59]. (**f**) The flexible deformable hydrofoils powered by SSC [60]. (**g**) The driving mechanism of Van der Geest et al.’s soft robotic green sea turtle [52].

**Figure 7 biomimetics-10-00272-f007:**
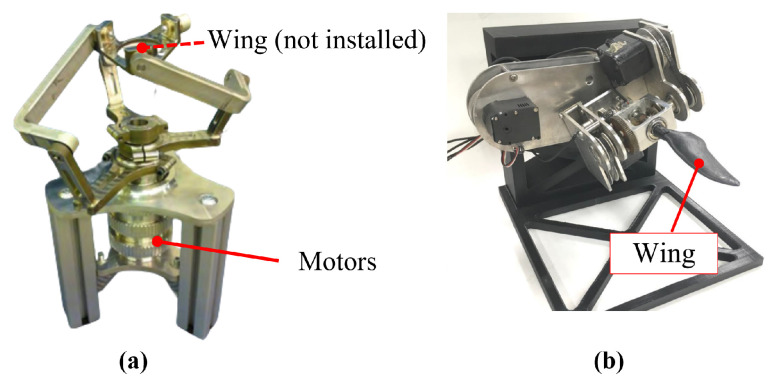
Penguin-inspired driving mechanisms: (**a**) the three-DOF spherical joint [61]; (**b**) the three-DOF robotic penguin wing [47].

**Figure 9 biomimetics-10-00272-f009:**
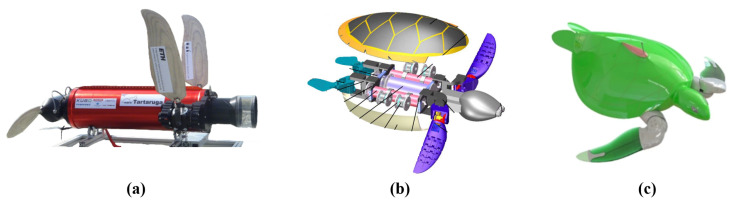
Sea turtle robots with 3-DOF hydrofoil: (**a**) ETH’s Naro-tartaruga [86]. (**b**) The robotic sea turtle with variable-stiffness fins [59]. (**c**) Van der Geest et al.’s soft robotic green sea turtle [52].

**Figure 10 biomimetics-10-00272-f010:**
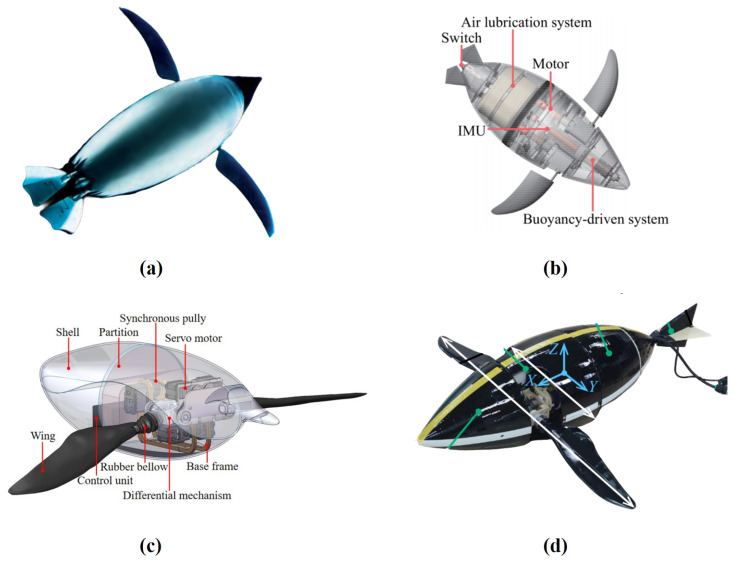
Penguin-inspired robots: (**a**) FESTO’s AquaPenguin [87]. (**b**) The penguin-inspired swimming robot with an air lubrication system [88]. (**c**) The 1:1 penguin-like robot with a 3-DOF wing [81]. (**d**) The agile robotic penguin from Tokyo Tech [89].

**Table 1 biomimetics-10-00272-t001:** Biomimetic underwater robot based on hydrofoil propulsion.

Name of Robot	Year	Biomimetic Object	DOF of Hydrofoil	Prototype Implemented	Specification
Naro-tartaruga [86]	2013	Sea Turtle	3	Yes	Length: 1 m
					Weight: 75 Kg
					Maximum Speed: 2 m/s
					Yaw rate: N/A
Turtle-inspired robot [59]	2022	Sea Turtle	3	No	N/A
Soft robotic green sea turtle [52]	2023	Sea Turtle	3	Yes	Length: 610 mm
					Weight: N/A
					Maximum Speed: 0.71 m/s
					Yaw Rate: N/A
AquaPenguin [87]	2009	Penguin	2	Yes	Length: 0.77 m
					Weight: 9.6 Kg
					Maximum Speed: 5 Km/h
					Yaw Rate: N/A
Penguin-inspired swimming robot [88]	2022	Penguin	2	Yes	Length: 0.71 m
					Weight: 19.4 Kg
					Maximum Speed: 0.56 m/s
					Yaw Rate: 72.6°/s
Penguin-like robot [81]	2023	Penguin	3	No	N/A
Agile robotic penguin [89]	2024	Penguin	2	Yes	Length: 886 mm
					Weight: 9.53 Kg
					Maximum Speed: 1.8 m/s
					Yaw rate: 92°/s

## Data Availability

Data are contained within the article.

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
