# Peer review of "Biomimetic Hydrofoil Propulsion: Harnessing the Propulsive Capabilities of Sea Turtles and Penguins for Robotics"

_biomimetics, 2025, doi:10.3390/biomimetics10050272_

Round 1
Reviewer 1 Report
Comments and Suggestions for Authors
Detailed as attachment

Author Response
Thank you for your review. We have revised the manuscript based on your comments. Please refer to the PDF for details of the changes.

Reviewer 2 Report
Comments and Suggestions for Authors
The article is devoted to a deep study of the principles and mechanisms of operation of hydrofoils of a green sea turtle and a penguin. General advantages of this type of propulsion in comparison with other known types of underwater propulsion in wildlife are noted. All the considered methods are characterized by low acoustic noise generated by movement, which makes them the most attractive. The advantages of the proposed principle are convincingly substantiated on the basis that, unlike other methods, this method has a number of valuable advantages. In particular, compared to the rowing method, there is no long recovery phase, since the wing creates forward thrust both when moving up and down, so the recovery period is so short that it can be argued about continuous thrust. Also, this method is the most effective in terms of minimum energy costs. The method based on body bending may allow achieving high speeds, but it is characterized by insufficient stability when implemented in robotics, in addition, it is much more complex and expensive.
It is noted that the principle of sea turtle swimming was previously understood incorrectly, researchers assumed that the forelimbs of the sea turtle functioned as simple paddles, and the push was created by water resistance. However, subsequent observational experiments and theoretical analyses showed that sea turtles use a mechanism of movement based on the lifting force, where the push is a component of the hydrodynamic lifting force in the forward direction. Thus, the forelimbs of the sea turtle are not paddles, but rather aquatic wings. This is fundamentally different from freshwater turtles. The review provides a detailed picture of how, as the study of the movement of the turtle's wings deepened , the understanding of this process was refined, and the theory of four-phase movement was replaced by the theory of five-phase movement.
The penguin's wings also create forward thrust both when they are lowered and when they are raised.
It is noted that for a more successful understanding of these processes, it is necessary to take into account not only the trajectory of the wing and not only its rotation during movement, determined by the angle of rotation of the wing tips, but also the bend due to the flexibility of the wing. Therefore, it is proposed to use the method of tracking a number of characteristic points on the wing, which determine the positions of all conditionally flat fragments of the entire wing as a whole.
The similarities and differences in the movement of the wings of sea turtles and penguins are highlighted, and these features are linked to the different anatomical structure of the skeleton of the aquatic wings of these animals.
The article provides a critical analysis of various wing drive mechanisms in robots inspired by sea turtle and penguin wings. The article also provides an overview of specific robots that use this principle of movement. Most often, such robots imitate the shape and appearance of the animals that inspired their creators.
The authors of the article also refer to their earlier publication [61]. They report their work very modestly and briefly in line 531, and there is also a single illustration of this result, Figure 9(c).
In addition, the article contains a reference to the article by the first author of this article, reference [43], which shows that in 2020 this author worked in Tokyo, at the Tokyo Institute of Technology. It is clear that during this work, as part of a team in Tokyo, this author studied the wing movements of a sea turtle, and currently, in two scientific organizations in the city of Nanjing, this researcher is working on a robot that imitates the wing movements of a penguin. Thus, the combination of two topics in one review article is dictated by the following considerations: firstly, the historical and current interests of the first author, secondly, the desire to write a review article on the accumulated materials due to this interest, and thirdly, simply the desire to publish the article in a highly rated journal.
The resulting article will certainly be of great interest to many readers, as it provides a broad and in-depth overview of the problems in general, and even the history of the study of the movement of the aquatic wings of the sea turtle and penguin.
But the article has some obvious shortcomings.
Firstly, the article is actually a purely review article, there is nothing new or original in it that has not been published before. It looks like an introduction to the dissertation of the first author. For this reason, the article does not have any new conclusions or new proposals based on them at the end. The indication that the task is complex is important, that there are many different approaches to solving it, this is not an original conclusion. In essence, the article calls for continuing research and doing it especially thoroughly and fully. Nothing more.
Secondly, if the authors write about their results, then they should not refer to them as if they were someone else's results and that the authors had nothing to do with them.
Thirdly, readers are interested to know that if the team publishing the article has its own results in this area, which have already been published earlier, in this case, in 2023, then how has this team of authors progressed in these studies, what new results have they obtained, since enough time has passed since 2023 for these results to appear. If this team has stopped research in this area, then writing a review is somewhat inappropriate, and if all the new results have already been published in another article, then it is also necessary to contrast that the authors have not left anything new from the results of their own research for this publication.
Therefore, despite the fact that the entire introductory part of this review is very well written and reads with interest, the reader experiences great disappointment when it turns out that, in addition to the introductory part offering an overview of other people's results, the article contains only a two-line reference to his own work, followed by conclusions that are, in fact, quite trivial.
For this reason, it is suggested that either the authors should strengthen the description of their own results by reporting after the reference to the article [61] what additional results were obtained after the creation of this robot, since it has apparently already been tested, used, some characteristics were measured, at what speed it can swim, how it can maneuver, and so on. Or the article should slightly correct the abstract and introduction, since in fact the article is only a review of already published articles, so that it basically does not report anything new, except for a list of other people's results, accompanied by illustrations from other people's articles.
The article definitely needs some revision. In the abstract, instead of stating that the article begins with a review, it should be stated that the article is a review. The article does not review several types of robots, but simply provides references and illustrations about these several types of robots. In this regard, other people's illustrations, such as Figure 1, Figure 4 are certainly and unconditionally superfluous, and many other other people's illustrations are not necessary.
Author Response

(The authors gave the same response as above.)

Reviewer 3 Report
Comments and Suggestions for Authors
The article is devoted to a topic of fundamental and applied interest. However, not only does it not contain new original results, but it also cannot claim to be any acceptable review considering the mechanisms used to move in a continuous environment which may be of interest to bionics. At a minimum, such a review should contain information about Lighthill's fish swimming model and its recent modifications. In the present form, I consider the publication of the material unjustified.
Comments on the Quality of English LanguageNo comments
Author Response

(The authors gave the same response as above.)

Round 2
Reviewer 3 Report
Comments and Suggestions for Authors
Now the publishing of this paper in the present form is possible.